# Enhancing Unsupervised Sentence Embeddings via Knowledge-Driven Data Augmentation and Gaussian-Decayed Contrastive Learning

## Abstract

Recently, using large language models (LLMs) for data augmentation has led to considerable improvements in unsupervised sentence embedding models. However, existing methods encounter two primary challenges: limited data diversity and high data noise. Current approaches often neglect fine-grained knowledge, such as entities and quantities, leading to insufficient diversity. Additionally, unsupervised data frequently lacks discriminative information, and the generated synthetic samples may introduce noise. In this paper, we propose a pipeline-based data augmentation method via LLMs and introduce the Gaussian-decayed gradient-assisted Contrastive Sentence Embedding (GCSE) model to enhance unsupervised sentence embeddings. To tackle the issue of low data diversity, our pipeline utilizes knowledge graphs (KGs) to extract entities and quantities, enabling LLMs to generate more diverse, knowledge-enriched samples. To address high data noise, the GCSE model uses a Gaussian-decayed function to limit the impact of false hard negative samples, enhancing the model's discriminative capability. Experimental results show that our approach achieves state-of-the-art performance in semantic textual similarity (STS) tasks, using fewer data samples and smaller LLMs, demonstrating its efficiency and robustness across various models.

## 1 Introduction

Sentence representation learning, a fundamental task in natural language processing (NLP), aims to produce accurate sentence embeddings, thereby improving performance in downstream tasks such as semantic inference (Reimers & Gurevych, 2019), retrieval (Thakur et al., 2021; Wang et al., 2022a), and question answering (Sen et al., 2020). To enhance computational efficiency and reduce labor costs, unsupervised sentence embedding methods based on contrastive learning, such as SimCSE (Gao et al., 2021) and ESimCSE (Wu et al., 2022c), have emerged as highly effective paradigms. In general, contrastive learning methods operate on the principle that effective sentence embeddings should pull similar sentences closer while pushing dissimilar ones further apart. The performance of unsupervised contrastive learning methods largely depend on the quantity and quality of the samples (Chen et al., 2022), making it crucial to develop strategies that effectively improve both.

Previous studies mainly focused on increasing the number of samples using rule-based word modifications (Wang & Dou, 2023; Wu et al., 2022c) or feature sampling and perturbation techniques (Xu et al., 2023; Chuang et al., 2022a). Recent studies (Zhang et al., 2023; Wang et al., 2024a) use either few-shot manually constructed samples or zero-shot generalized refactoring instructions to create prompts that guide large language models (LLMs) in generating new samples from original sentences, increasing both the quantity and quality of the data. Although these methods have achieved commendable performance, two limitations remain:

**Low Data Diversity.** Diverse data samples in sentence representation learning should contain varied expressions of the same knowledge. However, existing approaches often struggle to distinguish fine-grained semantic knowledge like entities and quantities in the context. Traditional methods modify sentences using limited patterns without considering fine-grained knowledge, restricting their effectiveness in enhancing sample diversity. Recent LLM-based methods like Wang et al. (2024b),

SynCSE (Zhang et al., 2023) and MultiCSR (Wang et al., 2024a), adjust topic and entailment categories in prompts to guide the model in generating varied samples. These methods focus on the global context but lack precise control over the knowledge in the samples. Consequently, the diversity of generated samples is constrained by the probability distributions of LLMs, resulting in unpredictable data quality.

**High Data Noise.** Unsupervised sentence representation learning often suffers from data noise caused by confusing negative samples, which mainly arise from two sources. First, traditional methods generate datasets by duplicating samples to create positive instances, leading to negatives with similar surface-level semantics that affect the model's performance (Miao et al., 2023; Zhou et al., 2022). Second, in data synthesis, differences in semantic distributions can cause the LLM's criteria for distinguishing between positive and negative samples to misalign with the target domain, introducing additional noise (Huang et al., 2023; Poerner & Schütze, 2019). The existing MultiCSR method attempts to remove noisy samples using linear programming, but this can eliminate potentially valuable samples and reduce data diversity. Figure 1 compares various baselines on the STS-Benchmark development set. The results show that the prediction of false positives outnumber false negatives, and data synthesis in SynCSE increases false negatives, further supporting the above analysis.

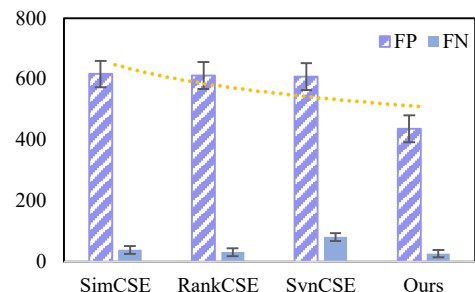

Figure 1: Comparison of false positives (FP) and negatives (FN). Both the predicted scores and labels are normalized (see details in Appendix I), where positives have a score greater than the label, while negatives lower than the label. False samples are identified when the root mean square error (RMSE) between the prediction and the label exceeds 0.2.

In this paper, we propose a pipeline-based data augmentation method using LLMs and introduce the Gaussian-decayed gradient-assisted Contrastive Sentence Embedding (GCSE) model to improve the performance of unsupervised sentence embedding methods. To address the issue of *low data diversity*, we begin by extracting entities and quantities from the data samples and constructing a knowledge graph (KG) with the extracted data. Next, we create

| Methods | Synthesis Approach | Use Knowledge | Denoise |
|---------|-------------------|---------------|---------|
| SynCSE | Few-shot Synthesis | No | No |
| MultiCSR | Zero-shot Synthesis | No | Yes |
| Ours | Zero-shot Synthesis | Yes | Yes |

Table 1: Comparison of our methods and related LLM-based methods.

a sentence construction prompt using the extracted knowledge to guide LLM in generating more diverse positive samples. To tackle *high data noise*, we employ an evaluation model to annotate the synthesized data and initially filter out false samples. However, this procedure is ineffective in filtering out false negatives with similar surface-level semantics. To balance sample diversity while minimizing the impact of noise from false negatives, we aim to align all hard negatives with the distribution of the evaluation model in the initial training step. Then, we leverage other in-batch negative samples to optimize the semantic space. Therefore, we propose the GCSE model that employs a Gaussian-decayed function to calculate the prediction distinctions between GCSE and the evaluation model. It first declines the gradients of hard negatives. As training progresses, the gradient weights for hard negatives that diverge farther from the evaluation model's distribution progressively recover. This function helps prevent false negatives from being pushed further away in the semantic space, leading to a more uniform distribution. We highlight the key innovations of our approach in Table 1: (i) We are the first to incorporate fine-grained knowledge for sample synthesis in LLM-based methods. (ii) Unlike MultiCSR's denoising approach, our method retains more false samples for training rather than discarding them. (iii) Our data selection strategy focuses on domain-specific samples, using a local LLM with fewer samples for synthesis, leading to improved performance. Experimental results demonstrate the efficiency of our model, outperforming previous best methods in average scores for semantic textual similarity (STS) tasks by 1.05% with BERT-base, 1.62% with BERT-large, 0.49% with RoBERTa-base, and 1.50% with RoBERTa-large.

In summary, our contributions are as follows: (1) *New method.* We introduce a pipeline-based data augmentation method using LLM for few-shot domain data and propose a Gaussian-decayed

Figure 2: The overall workflow of our method.

gradient-assisted Contrastive Sentence Embedding (GCSE) model to reduce data noise. (2) *New perspective.* To the best of our knowledge, we are the first to explore combining knowledge graphs with LLM to synthesize data, enhancing fine-grained sentence representation learning by generating diverse positive and negative samples. (3) *State-of-the-art performance.* Experimental results demonstrate that our method achieves superior performance on STS tasks while using fewer samples for data synthesis with smaller LLM parameters.

## 2 RELATED WORK

Early work on sentence embeddings builds on the distributional hypothesis, predicting surrounding sentences (Kiros et al., 2015; Logeswaran & Lee, 2018; Hill et al., 2016) or extending the word2vec framework (Mikolov et al., 2013) with n-gram embeddings (Pagliardini et al., 2018). Post-processing techniques like BERT-flow (Li et al., 2020) and BERT-whitening (Su et al., 2021) address the anisotropy issue in pre-trained language models (PLMs), and more recent methods focus on generative approaches (Wang et al., 2021; Wu & Zhao, 2022) and regularizing embeddings to prevent representation degeneration (Huang et al., 2021). Recently, contrastive learning approaches have become prominent, using various augmentation methods to derive different views of the same sentence (Zhang et al., 2020; Giorgi et al., 2021; Kim et al., 2021; Gao et al., 2021). Among these, Sim-CSE uses dropout as a simple augmentation and achieves strong results in unsupervised STS tasks, inspiring further approaches like ArcCSE (Zhang et al., 2022), DiffCSE (Chuang et al., 2022a), GS-InfoNCE (Wu et al., 2022b), and RankCSE (Liu et al., 2023).

With the advent of LLM (OpenAI, 2023; Bai et al., 2023; Touvron et al., 2023), some works attempt to utilize LLM for sentence representation learning. For example, Ni et al. (2022) uses T5 with mean pooling to obtain a sentence embedding model by fine-tuning on a large-scale NLI corpus; Cheng et al. (2023) uses prompt learning to measure the semantic similarity of sentence pairs; Springer et al. (2024) employs sentence repetition to enhance the capacity for sentence representation; AoE (Li & Li, 2024a) optimize angle differences for improving supervised text embedding; and BeLLM (Li & Li, 2024b) designs a Siamese structure for learning sentence embeddings.

## 3 METHODOLOGY

In this section, we present the data augmentation pipeline via LLM and the specific structure of the GCSE. As shown in Figure 2, we start by using a data augmentation pipeline to synthesize new samples from the source data, and then train our model with the filtered synthetic data.

### 3.1 DATA AUGMENTATION

In the data augmentation pipeline, we utilize both domain data and partial general data to balance domain-specific relevance and general-domain applicability. We start by extracting knowledge from the source data and then synthesize new data for our model training. The detailed structure of the pipeline is shown in Figure 3.

**Knowledge Extraction and Integration.** The variety and relationships between samples directly impact model performance in sentence representation learning. A major challenge with existing LLM-based data synthesis methods is the limited diversity they generate for each short text. To trade off the low diversity of the generated samples with their relevance to the domain semantic space, we first design an extraction prompt to obtain entities and quantities from the given data.

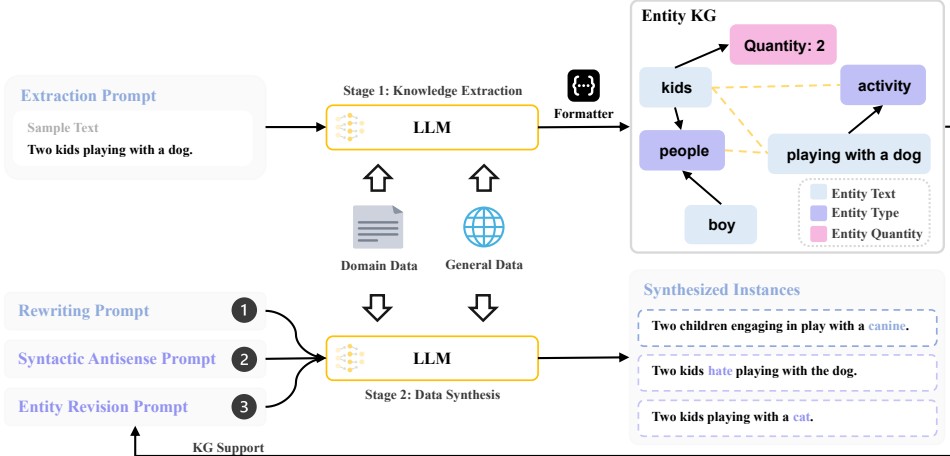

Figure 3: The pipeline of knowledge extraction and data synthesis, where the solid black arrows in the Entity KG are hard edges, and dotted yellow lines are soft edges.

Formally, we denote the extraction prompt as $\mathcal{P}_e$, and LLM $\mathcal{L}$, suppose we finally extract instances with $d$ sample number, the knowledge set $\mathcal{K}_i = \{k_{i1}, \ldots, k_{in}\}$ of each instance $x_i$ is computed in Equation 1, where $t_j$, $c_j$ and $q_j$ represent the entity text, entity type and quantity of $k_i$. $n_i$ is the size of $\mathcal{K}_i$, and $\mathcal{F}(\cdot)$ is the formatting function that convert text to triplet. Next, we integrate all knowledge by establishing an entity knowledge graph $\mathcal{G} = \langle V, E \rangle$, where the node set $V$ contains all the $\langle t, c, q \rangle$ from $\mathcal{K}$:

$$\mathcal{K} = \bigcup_{i=1}^{d} \mathcal{F}([\mathcal{P}_e; x_i], \mathcal{L}) = \bigcup_{i=1}^{d}\{\langle t_{ij}, c_{ij}, q_{ij}\rangle \mid j \in [1, n_i]\}, \tag{1}$$

$$V = \{t_{ij}, c_{ij}, q_{ij} \mid i \in [1, d]; j \in [1, n_i]\}. \tag{2}$$

The edges $E$ consist of hard edges $E_r$ and soft edges $E_s$. As shown in Equations 3 and 4, $E_r$ represents the relationship between the entity text, type and quantity of each $k \in \mathcal{K}$, and $E_s$ indicates the relationship between entity text in $k_{ij}$ and other entity text or type in the same instance $x_i$.

$$E_r = \{(t_{ij}, c_{ij}) \cup (t_{ij}, q_{ij}) \mid i \in [1, d]; j \in [1, n_i]\}, \tag{3}$$

$$E_s = \bigcup_{i=1}^{d}\{(t_{ij}, t_{ik}), (t_{ij}, c_{il}) \mid k, l \neq j; j, k, l \in [1, n_i]\}. \tag{4}$$

By defining hard and soft edges, we can more efficiently identify and replace entity nodes near the current node, improving the correlation between the synthesized instance and the source instance.

**Data Synthesis via LLM.** Empirical evidence and model performance on standard datasets show that sentence embedding models struggle more with accurately identifying negative samples than positives (Chuang et al., 2022a; Miao et al., 2023). In the contrastive learning methods, the model acquires sentence embedding representation by calculating the distance between sentence-pairs. It aims to minimize the spatial distance between positive pairs and increase the spatial distance between negative pairs. Thus, it is essential to obtain negative samples that closely resemble the source instance in surface-level features, while positive samples should have diverse representations but still convey the same meaning as the source instance.

In this study, we use LLM to generate positive samples through a rewrite prompt. We also focus on the impact of variations in entities and quantities within the samples. Negative samples are generated by the LLM at both the syntactic and fine-grained knowledge levels. The data synthesis prompts are divided into three main types: (1) Rewriting prompt, (2) Syntactic antisense prompt, and (3) Entity revision prompt. The first type is used to create positive samples, while the second and third types are used to create negative samples at the syntactic and knowledge levels, respectively.

The "rewriting prompt" can be classified into three forms: directly requesting LLM to generate a new sentence instance using the "rewrite" instruction, creating the preceding part of the sentence instance, and generating based on the knowledge set of the instance. As the diversity of synthetic samples increases, the likelihood of generating false positives also rises. To address this, the next section involves scoring the generated samples using an evaluation model. The "syntactic antisense prompt" aims to modify the semantics to create a contradiction at the syntactic level. Such as transforming it into a positive/negative statement using explicit positive/negative words, or by expressing a contrary sentiment. This is an initial approach to synthesizing negative samples that preserves a strong coherence with the source instance in terms of sequence structure. However, it is deficient in generation diversity. To alleviate the issue, the "entity revision prompt" aims to enhance text diversity by replacing the entity text and quantity compared to the source instance. Simultaneously, to ensure the semantic relevance between the synthetic samples and the source instance, replacement entities are selected by searching for neighboring nodes on entity KG. We define $\mathcal{T}(\cdot)$ as the search function, and the replacement entity of $t_{ij}$ are computed as:

$$\mathcal{T}_r(t_{ij}) = \{t_{ip} \mid (t_{ij}, c_{ik}) \in E_r \wedge (t_{ip}, c_{ik}) \in E_r\}, \tag{5}$$

$$\mathcal{T}_s(t_{ij}) = \{t_{ip} \mid (t_{ij}, t_{ip}) \in E_s\}, \tag{6}$$

$$\mathcal{T}_p(t_{ij}) = \{t_{ip} \mid t_{ik} \in \mathcal{T}_s(t_{ij}) \cap \mathcal{T}_s(t_{ip}) \wedge t_{ip} \in \mathcal{T}_r(t_{ij})\}, \tag{7}$$

$$\mathcal{T}(t_{ij}) = \mathcal{T}_r(t_{ij}) \cup \mathcal{T}_p(t_{ij}), \tag{8}$$

where the function $\mathcal{T}_r(\cdot)$ is used to search for entities that have a hard edge with the current entity, and $\mathcal{T}_s(\cdot)$ is used to search for entities that have a soft edge with the current entity. $\mathcal{T}_p(\cdot)$ aims to search for $t_{ip}$, that is of the same type as $t_{ij}$, and they both have soft edges with another in-context entity $t_{ik}$. Finally, the replacement entity can be randomly selected from the result of the search function $\mathcal{T}(t_{ij})$. Compared to randomly replacing entities, our strategy enhances the semantic relevance between the generated sample and the source instance.

## 3.2 MODEL TRAINING

The training process of our model consists of two stages. First, we combine general and domain-specific data to train an evaluation model using standard unsupervised contrastive learning. This improves the uniformity of sentence embeddings in general scenarios and reduces the impact of semantic distribution limitations in the synthesized data, enhancing model robustness. Then, we freeze the evaluation model to filter synthetic data and help the GCSE model eliminate false hard negative sample noise.

**General Contrastive Learning.** In the first stage, we follow the formulation of SimCSE (Gao et al., 2021) to train the evaluation model. Formally, we define the encoder of the evaluation model as $E'$, each unlabeled sentence instance as $x_i$, and its positive sample as $x_i^+ = x_i$. The representation of each instance is denoted as $\mathbf{h}' = \mathcal{F}_{E'}(x)$, the representations of $x_i$ and $x_i^+$ are computed as $\mathbf{h}'_i$ and $\mathbf{h}_i'^+$, respectively. Since the dropout mask in $E'$ is random, $\mathbf{h}'_i$ and $\mathbf{h}_i'^+$ are computed with the same input but with slightly different results. Then, the loss of evaluation model is defined as:

$$-\log \frac{e^{\text{sim}(\mathbf{h}'_i, \mathbf{h}_i'^+)/\tau}}{\sum_{j=1}^N e^{\text{sim}(\mathbf{h}'_i, \mathbf{h}_j'^+)/\tau}}, \tag{9}$$

where $N$ represents the size of each mini-batch, $\tau$ is a temperature hyperparameter, and $\text{sim}(\cdot)$ is the cosine similarity function.

**Denoising Training.** In the second stage, we adopt a copy of the evaluation model as the backbone of GCSE and continue training on synthesized data. In this stage, each input is set as a triplet $(x_i, x_i^+, x_i^-)$, where $x_i^+$ and $x_i^-$ stand for the positive and negative samples of $x_i$, respectively. Nevertheless, the synthesized data contains many potential false positive and false negative samples, necessitating the implementation of a filtering process. We use the frozen evaluation model to initially correct these inaccurate samples and build the ultimate triplet dataset. Let $\mathcal{S}(x_i) = \{\hat{x}_{i1}, \ldots \hat{x}_{im}\}$ denotes the synthetic data set of $x_i$, where $m$ is the size of the set, and $x_i^+, x_i^-$ are calculated as:

$$x_i^+ = \begin{cases} \hat{x}_{ij}, & \text{sim}(\mathbf{h}'_i, \hat{\mathbf{h}}'_{ij}) \geq \alpha, j \in [1, m] \\ x_i, & \text{else} \end{cases}, \tag{10}$$

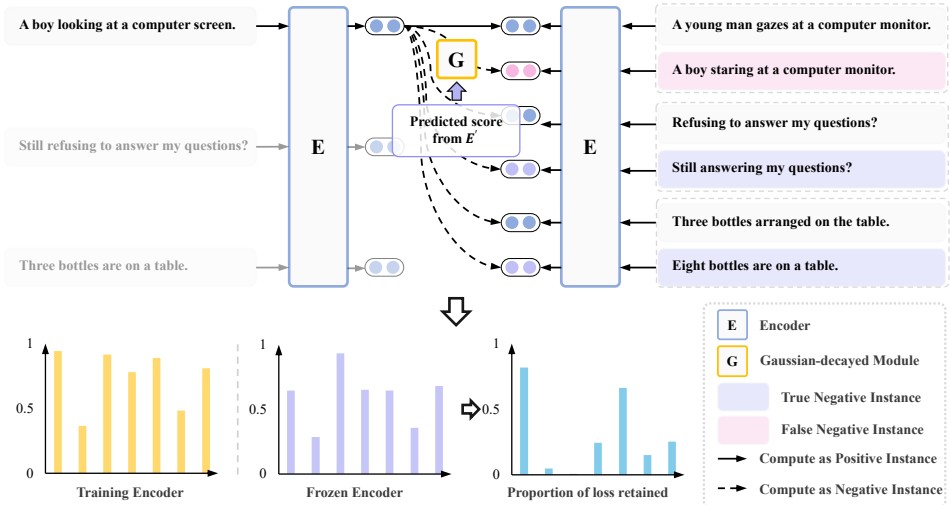

Figure 4: In-batch training with Gaussian-decayed on GCSE.

$$x_i^- = \begin{cases} \hat{x}_{ij}, & \text{sim}(\mathbf{h}_i', \hat{\mathbf{h}}_{ij}') \leq \beta, j \in [1, m] \\ x_k, & k \in [1, N], k \neq i \end{cases}, \tag{11}$$

where $\alpha$, $\beta$ are the threshold for positives and negatives, respectively. $x_k$ denotes a randomly selected instance from in-batch data. We can set a high value for $\alpha$ to reduce false positive samples. However, filtering out false negatives in synthetic data is more challenging. In theory, smaller $\beta$ can reduce more false negatives, but samples with low similarity to the source instance are easy to distinguish due to significant surface-level differences. As a result, training on these samples does not effectively improve the model's ability to distinguish fine-grained false positives. Therefore, we opt for a higher value of $\beta$. During training, we use a Gaussian-decayed function to align the distances of hard negative samples between the GCSE encoder $E$ and the frozen encoder $E'$. As shown in Figure 4, for each mini-batch of triplet inputs, both $E$ and $E'$ compute similarity scores for the negative samples and their corresponding source instances. The loss for each instance in GCSE is defined as:

$$-\log \frac{e^{\text{sim}(\mathbf{h}_i, \mathbf{h}_i^+)/\tau}}{\sum_{j=1}^N e^{\text{sim}(\mathbf{h}_i, \mathbf{h}_j^+)/\tau} + \sum_{\substack{j=1 \\ j \neq i}}^N e^{\text{sim}(\mathbf{h}_i, \mathbf{h}_j^-)/\tau} + G(s_i, s_i', \tau, \sigma)}, \tag{12}$$

$$G(s_i, s_i', \tau, \sigma) = s_i \left(1 - e^{-\frac{(s_i - s_i')^2 \tau^2}{2\sigma^2}}\right), \tag{13}$$

where $s_i = \text{sim}(\mathbf{h}_i, \mathbf{h}_i^-)$, $s_i' = \text{sim}(\mathbf{h}_i', \mathbf{h}_i'^-)$. $G(\cdot)$ is the Gaussian-decayed function, where the loss attenuation of the hard negative sample grows as the distance between $s_i$ and $s_i'$ decreases, and $\sigma$ is a hyperparameter that controls the width of $G(\cdot)$. This implies that when $E$ initially calculates the hard negative sample, it follows the spatial distribution of $E'$ as the "established guidelines" and uses other in-batch negative samples to further increase the spatial distance between negatives, effectively reducing the influence of false negatives. As training progresses, the spatial distribution of true hard negatives between $E$ and $E'$ will progressively increase, and its gradient will be restored.

## 4 EXPERIMENT

### 4.1 EXPERIMENT SETUP

**Training:** We utilize the subset of NLI dataset from Gao et al. (2021) as the general data, and use the training sets from STS-Benchmark (STS-B) (Cer et al., 2017) with 5.7k samples and SICK (Marelli et al., 2014) with 4.5k samples as the domain data for a fair comparison with related approaches.

To simulate the unsupervised scenario, we exclusively include unlabeled samples from the dataset. In this experiment, the ratio of sample numbers between domain data and general data was 1:3. We adopt ChatGLM3-6B (GLM et al., 2024), GLM4-9B (GLM et al., 2024) and ChatGPT (OpenAI, 2022) as LLMs for data synthesis, respectively. We choose BERT (Devlin et al., 2019) and RoBERTa (Liu et al., 2019) as the backbone models of GCSE. In the stage of Gaussian-decayed training on synthesized data, the filtering thresholds of $\alpha$ and $\beta$ are set as 0.9 and 0.75, respectively. The temperature of $\tau$ is set as 0.05, and the $\sigma$ of $G(\cdot)$ is set as 0.01. In the first stage training, the evaluation model is firstly trained on the unlabeled dataset of all general data and domain data. One copy instance of the evaluation model is then utilized as the pre-trained model for GCSE, while the original instance is set to be frozen to filter synthesized data and provide guidance for GCSE. In the second stage, GCSE is trained on the filtered synthesized data, and the sentence embedding is obtained from the last output hidden states of the first token.

**Evaluation:** To validate our method for sentence embeddings, we evaluated the model's performance on semantic textual similarity (STS) tasks, we use the standard evaluation method, measuring model performance with Spearman's correlation, and we adopt SentEval[1] (Conneau & Kiela, 2018) as the evaluation tool, which contains seven STS subsets: STS 2012-2016 (Agirre et al., 2012; 2013; 2014; 2015; 2016), the STS-Benchmark (Cer et al., 2017) and the SICK Relatedness (Marelli et al., 2014). To compare the ranking performance of our method on retrieval tasks, we evaluated the model using the MTEB benchmark (Muennighoff et al., 2023) with four reranking datasets: AskUbuntuDupQuestions (Lei et al., 2016), MindSmallReranking (Wu et al., 2020), SciDocsRR (Cohan et al., 2020) and StackOverflowDupQuestions (Liu et al., 2018), and follow the same settings of Zhang et al. (2023) by using Mean Average Precision (MAP) as the metric. Additionally, we compared the performance of our model with other methods on transfer tasks in SentEval to evaluate its applicability in Appendix C.

**Baselines:** We compare our method with mainstream unsupervised sentence embedding baselines: BERT-whitening (Su et al., 2021), SimCSE (Gao et al., 2021), DiffCSE (Chuang et al., 2022b), PromptBERT (Jiang et al., 2022), PCL (Wu et al., 2022a), CARDS (Wang et al., 2022b), DebCSE (Miao et al., 2023) and RankCSE (Liu et al., 2023). In addition, we further compare two baselines: SynCSE (Zhang et al., 2023) and MultiCSR (Wang et al., 2024a), which use LLM for data synthesizing in whole NLI datasets. To verify the effectiveness of our data synthesis method, we choose their results of using ChatGPT for comparison.

## 4.2 Main Results

**STS Tasks:** The overall results of the STS tasks are shown in Table 2. Our approach, utilizing synthetic samples from ChatGPT, achieves state-of-the-art performance across all backbones when compared to other unsupervised baselines. Even with synthetic samples from ChatGLM3-6B, our method still outperforms previous approaches on BERT-base, BERT-large, and RoBERTa-large. This highlights the applicability of our method, as it can be effectively applied to multiple models. Compared to the standard unsupervised SimCSE, Spearman's correlation of GCSE (ChatGLM3-6B) is improved by an average of 17.24% on the base models and 3.44% on the large models. On the strong baseline RankCSE, GCSE (ChatGLM3-6B) achieved a 1.36% improvement over its average performance, demonstrating the effectiveness of the LLM data synthesis process. Furthermore, we compare two baseline models: SynCSE and MultiCSR, both of which utilize LLM as the data synthesis model. We specifically analyze the results of using ChatGPT for both models and the results show that our approach outperforms both models in most cases. It should be noted that our method only utilizes 14% of the sample size compared to the other two methods that employ the entire NLI datasets. This demonstrates the effectiveness of our data synthesis strategy and domain-oriented sample selection strategy.

**Reranking Tasks:** Table 3 presents the MAP results of our approach and related baselines on the reranking benchmark, and all models are evaluated on the test sets of the reranking benchmark without using the training sets. The results indicate that various approaches exhibit varying performance on different datasets, which can be attributed to the distinct semantic distribution and evaluation scale of each dataset. Our GCSE outperforms SynCSE by 0.39% in average MAP score and achieves the

---

[1]https://github.com/facebookresearch/SentEval

| Model | Method | STS-12 | STS-13 | STS-14 | STS-15 | STS-16 | STS-B | SICK-R | Avg. |
|---|---|---|---|---|---|---|---|---|---|
| BERT-base | whitening† | 57.83 | 66.90 | 60.90 | 75.08 | 71.31 | 68.24 | 63.73 | 66.28 |
| | SimCSE† | 68.40 | 82.41 | 74.38 | 80.91 | 78.56 | 76.85 | 72.23 | 76.25 |
| | DiffCSE† | 72.28 | 84.43 | 76.47 | 83.90 | 80.54 | 80.59 | 71.23 | 78.49 |
| | PromptBERT♣ | 71.56 | 84.58 | 76.98 | 84.47 | 80.60 | 81.60 | 69.87 | 78.54 |
| | PCL♠ | 72.84 | 83.81 | 76.52 | 83.06 | 79.32 | 80.01 | 73.38 | 78.42 |
| | DebCSE† | 76.15 | 84.67 | 78.91 | 85.41 | 80.55 | 82.99 | 73.60 | 80.33 |
| | RankCSE♠ | 75.66 | **86.27** | 77.81 | 84.74 | 81.10 | 81.80 | 75.13 | 80.36 |
| | SynCSE (ChatGPT)* | 75.86 | 82.19 | 78.71 | **85.63** | 81.11 | 82.35 | _78.79_ | 80.66 |
| | MultiCSR (ChatGPT)♣ | 74.86 | 84.19 | 79.46 | 84.70 | 80.34 | 83.59 | **79.37** | 80.93 |
| | **GCSE (ChatGLM3-6B)** | 76.91 | _86.23_ | _80.49_ | _85.16_ | _81.45_ | 82.54 | 75.71 | 81.21 |
| | **GCSE (GLM4-9B)** | _78.19_ | 84.88 | 80.28 | 84.39 | **81.81** | **83.89** | 77.74 | _81.60_ |
| | **GCSE (ChatGPT)** | **78.20** | 85.90 | **81.17** | 84.88 | 81.44 | _83.56_ | 78.69 | **81.98** |
| BERT-large | SimCSE† | 70.88 | 84.16 | 76.43 | 84.50 | 79.76 | 79.26 | 73.88 | 78.41 |
| | PCL♠ | 74.87 | 86.11 | 78.29 | 85.65 | 80.52 | 81.62 | 73.94 | 80.14 |
| | DebCSE† | 76.82 | 86.36 | 79.81 | _85.80_ | 80.83 | 83.45 | 74.67 | 81.11 |
| | RankCSE♠ | 75.48 | 86.50 | 78.60 | 85.45 | 81.09 | 81.58 | 75.53 | 80.60 |
| | SynCSE (ChatGPT)* | 74.24 | 85.31 | 79.41 | 85.71 | **81.76** | 82.61 | _79.25_ | 81.18 |
| | **GCSE (ChatGLM3-6B)** | _76.99_ | **87.34** | 80.88 | 85.47 | 80.55 | 82.97 | 75.68 | 81.41 |
| | **GCSE (GLM4-9B)** | 76.94 | 86.69 | 81.16 | 85.53 | 81.44 | **84.47** | 78.88 | _82.16_ |
| | **GCSE (ChatGPT)** | **78.70** | _87.30_ | **81.94** | **86.10** | 81.60 | _84.08_ | **79.86** | **82.80** |
| RoBERTa-base | whitening† | 46.99 | 63.24 | 57.23 | 71.36 | 68.99 | 61.36 | 62.91 | 61.73 |
| | SimCSE† | 70.16 | 81.77 | 73.24 | 81.36 | 80.65 | 80.22 | 68.56 | 76.57 |
| | DiffCSE† | 70.05 | 83.43 | 75.49 | 82.81 | 82.12 | 82.38 | 71.19 | 78.21 |
| | PromptRoBERTa♣ | 73.94 | 84.74 | 77.28 | 84.99 | 81.74 | 81.88 | 69.50 | 79.15 |
| | PCL♠ | 71.13 | 82.38 | 75.40 | 83.07 | 81.98 | 81.63 | 69.72 | 77.90 |
| | DebCSE† | 74.29 | _85.54_ | 79.46 | _85.68_ | 81.20 | 83.96 | 74.04 | 80.60 |
| | RankCSE♠ | 73.20 | **85.95** | 77.17 | 84.82 | 82.58 | 83.08 | 71.88 | 79.81 |
| | SynCSE (ChatGPT)†† | 74.61 | 83.76 | 77.89 | 85.09 | 82.28 | 82.71 | 78.88 | 80.75 |
| | MultiCSR (ChatGPT)♣ | 75.61 | 84.33 | 80.10 | 84.98 | 82.13 | 84.54 | **79.67** | 81.62 |
| | **GCSE (ChatGLM3-6B)** | 76.06 | 85.30 | _80.38_ | 85.28 | **83.26** | 84.07 | 74.55 | 81.27 |
| | **GCSE (GLM4-9B)** | _77.13_ | 85.05 | _80.25_ | 84.89 | _83.08_ | **84.78** | 76.63 | _81.69_ |
| | **GCSE (ChatGPT)** | **78.03** | 83.79 | **80.61** | **86.28** | 82.76 | _84.31_ | _79.01_ | **82.11** |
| RoBERTa-large | SimCSE† | 72.86 | 83.99 | 75.62 | 84.77 | 81.80 | 81.98 | 71.26 | 78.90 |
| | PCL♠ | 74.08 | 84.36 | 76.42 | 85.49 | 81.76 | 82.79 | 71.51 | 79.49 |
| | DebCSE† | 77.68 | 87.17 | 80.53 | 85.90 | 83.57 | 85.36 | 73.89 | 82.01 |
| | RankCSE♠ | 73.20 | 85.83 | 78.00 | 85.63 | 82.67 | 84.19 | 73.64 | 80.45 |
| | SynCSE (ChatGPT)†† | 75.45 | 85.01 | 80.28 | 86.55 | 83.95 | 84.49 | **80.61** | 82.33 |
| | **GCSE (ChatGLM3-6B)** | **78.24** | _87.24_ | 81.93 | 86.80 | 83.52 | 85.08 | 76.70 | 82.79 |
| | **GCSE (GLM4-9B)** | 77.18 | 86.72 | **82.62** | 85.89 | 83.97 | 85.75 | 77.97 | 82.87 |
| | **GCSE (ChatGPT)** | _77.76_ | **87.45** | **82.62** | **88.38** | **84.43** | **86.08** | _80.09_ | **83.83** |

Table 2: Comparison of Spearman's correlation results on STS tasks, where the value highlighted in bold is the best value, and the value underlined is the second-best value. "†": results from Miao et al. (2023), "♣": results from Wang et al. (2024a), "♠": results from Liu et al. (2023), "††": results from Zhang et al. (2023). "*": we reproduce the results with the officially released corpus from Zhang et al. (2023). GCSE has significant differences with all comparable baselines on the t-test ($p<0.5\%$).

best results in all backbone models, demonstrating the efficacy of our approach in enhancing the precision of unsupervised ranking tasks.

## 4.3 ANALYSIS

**Ablation Studies:** We analyze the impact of each module or strategy in GCSE and report the results in Table 4. First, "w/o stage-2" refers to the results obtained without training in the second stage. This leads to a significant decrease in performance compared to the default model, which is the performance of the evaluation model and is similar to the conventional unsupervised SimCSE. Then, "w randomly" refers to the direct use of the instance itself as a positive sample in the combination dataset of domain and general data, while randomly selecting a negative instance from the dataset. We can observe that its performance in this case is even worse than the evaluation model. This demonstrates that the diversity of positive samples and the quality of negative samples significantly impact the performance of the model. "w/o filtering" indicates the results of training by skipping evaluation model filtering and directly using the data synthesized by LLM. The results show that the performance of the model is significantly affected when false positive and negative samples are introduced without filtering. We investigate the impact of the Gaussian-decayed function by removing it, and the results are shown in "w/o decay". We can observe that the default model performs better overall than when the Gaussian-decayed function is removed, indicating that it can filter out potential false negative sample noise. Finally, we analyze the necessity of including general data and domain data in "w/o general" and "w/o domain" respectively. It can be observed that

removing either of them results in a decline in performance, which indicates the significance of domain data and the essentiality of general data in our method.

**Analysis of entities and quantities awareness:** We analyze GCSE awareness of entities and quantities by constructing a dataset using the data synthesis method in Section 3.1 on the STS-Benchmark development set. Then, the similarity scores of each triplet in the dataset are annotated by two supervised pretrained models: "sup-simcse-bert-large" and "sup-simcse-roberta-large". The final label is the average score of the similarity calculated by both models. We evaluate Spearman's correlation scores of GCSE and the other three strong baselines on the backbone of the BERT-base

| Method | Spearman's |
|---|---|
| unsup-SimCSE | 75.59 |
| RankCSE | 79.74 |
| SynCSE (ChatGPT) | 91.58 |
| GCSE (ChatGLM3-6B) | **93.77** |

Table 5: Comparison of Spearman's correlation results on the synthetic data of the STS-Benchmark development set.

model, and the results are shown in Table 5. Our GCSE achieves the best result and outperforms RankCSE by 14.03%. In this case, both SynCSE and GCSE achieve significant improvements over methods without LLM. This might be due to the similarity of the semantic representation space between the training set and the development set, both of which are synthesized via LLM. Nevertheless, GCSE shows a notable enhancement in performance of 2.19% compared to SynCSE, demonstrating that its understanding of the entities and quantities in sentences has enhanced to a certain degree.

**Impact on the ratio between domain and general data:** Figure 5 presents the trend of the GCSE Spearman's correlation result as the proportion of general data introduced increases, where "d" represents that only using the domain data. The results show that adding a certain amount of general data improves performance on STS tasks. However, when the size of general data exceeds three times that of domain data, performance starts to decline. This suggests that incorporating a moderate amount of external data enhances the uniformity of sentence embeddings. But as the out-of-domain data grows, the influence of domain-specific data on training weakens. Overall, the results indicate that domain data improves the model's ability to represent target domain sentences, while general data helps with sentence embedding uniformity.

**Impact of the Gaussian-decayed:** To further investigate the effectiveness of the Gaussian-decayed function, we analyze the GCSE performance against the weight of $\sigma$ on the synthesized data, both with and without filtering. As shown in Figure 6, we use the synthesized data without filtering to evaluate the efficacy of the Gaussian-decayed function in eliminating false negative samples, and

| Model | Method | AskU. | Mindsmall | SciDocsRR | StackO. | Avg. |
|---|---|---|---|---|---|---|
| BERT-base | SimCSE | 51.89 | 28.68 | 67.88 | 39.60 | 47.01 |
| | PCL | 52.46 | 28.72 | 68.03 | **41.30** | 47.63 |
| | SynCSE (ChatGPT)* | 52.61 | **29.17** | 68.46 | 38.60 | 47.21 |
| | GCSE (ChatGLM3-6B) | **52.62** | 28.79 | **70.67** | 39.53 | **47.90** |
| BERT-large | SimCSE | 53.10 | 29.59 | 71.94 | 40.68 | 48.83 |
| | PCL | 52.03 | 29.11 | 70.30 | **42.33** | 48.44 |
| | SynCSE (ChatGPT)* | 53.24 | **30.09** | 71.45 | 39.24 | 48.50 |
| | GCSE (ChatGLM3-6B) | **53.40** | 29.43 | **73.04** | 39.68 | **48.89** |
| RoBERTa-base | SimCSE†† | 52.78 | 29.91 | 65.96 | 39.25 | 46.95 |
| | CARDS†† | 52.94 | 27.92 | 64.62 | **41.51** | 46.75 |
| | PCL†† | 51.85 | 27.92 | 64.70 | 41.18 | 46.41 |
| | SynCSE (ChatGPT)†† | 53.27 | **30.29** | 67.55 | 39.39 | 47.63 |
| | GCSE (ChatGLM3-6B) | **53.44** | 29.35 | **67.89** | 41.13 | **47.95** |
| RoBERTa-large | SimCSE†† | 55.10 | 29.23 | 68.54 | 42.56 | 48.86 |
| | CARDS†† | 53.83 | 29.07 | 68.26 | **43.24** | 48.60 |
| | PCL†† | 53.43 | 28.56 | 66.06 | 41.54 | 47.40 |
| | SynCSE (ChatGPT)†† | **55.48** | 30.27 | 70.85 | 40.00 | 49.15 |
| | GCSE (ChatGLM3-6B) | 54.05 | **30.30** | **71.23** | 41.65 | **49.31** |

Table 3: Comparison of Mean Average Precision (MAP) results on reranking tasks, where the value highlighted in bold is the best value, and the value underlined is the second-best value. "††": results from Zhang et al. (2023). "*": we reproduce the results with the officially released corpus from Zhang et al. (2023).

| Method | STS-12 | STS-13 | STS-14 | STS-15 | STS-16 | STS-B | SICK-R | Avg. |
|---|---|---|---|---|---|---|---|---|
| GCSE (ChatGLM3-6B) | **76.91** | **86.23** | **80.49** | 85.16 | **81.45** | 82.54 | 75.71 | **81.21** |
| w/o stage-2 | 71.85 | 83.65 | 76.84 | 83.37 | 78.74 | 79.10 | 71.69 | 77.89 |
| w randomly | 71.94 | 84.03 | 76.99 | 83.65 | 79.11 | 78.66 | 69.28 | 77.67 |
| w/o filtering | 74.65 | 83.54 | 77.39 | 83.27 | 79.97 | 79.66 | 74.27 | 78.96 |
| w/o decay | 76.15 | 85.83 | 79.77 | **85.19** | 80.72 | **82.59** | 75.55 | 80.83 |
| w/o general | 75.44 | 85.55 | 79.19 | 84.91 | 80.23 | 81.57 | 74.14 | 80.15 |
| w/o domain | 75.59 | 85.66 | 78.93 | 84.09 | 80.87 | 82.29 | **76.00** | 80.49 |

Table 4: Ablation studies of STS tasks on BERT-base. Other PLMs yield similar patterns to BERT-base.

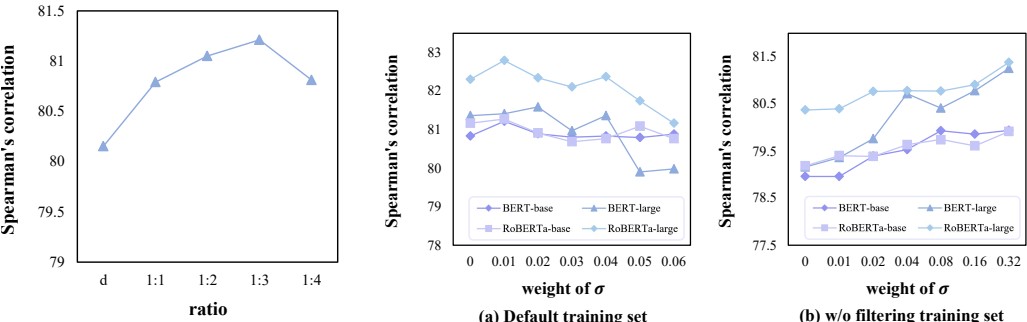

Figure 5: Spearman's correlation against the ratio of domain data to general data on the STS tasks.

Figure 6: Spearman's correlation against the weight of the Gaussian-decayed on the STS tasks.

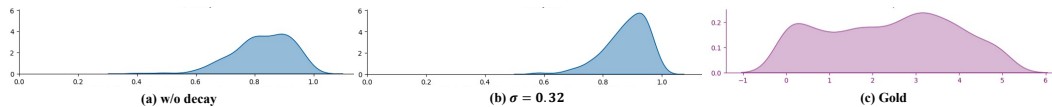

Figure 7: Density plots of the STS-Benchmark development set with labels $\geq 4$, which is evaluated by GCSE with different $\sigma$ weights. (c) is the density plot of gold labels.

results are presented in Figure 6 (b). It is clear that the model's performance improves as the weight of $\sigma$ grows. This suggests that a greater $\sigma$ weight enhances the model's effectiveness in mitigating the impact of false negative samples. It is important to acknowledge that a higher $\sigma$ does not necessarily indicate better performance. As shown in Figure 6 (a), an increase in $\sigma$ at the initial stage contributes to enhancing the model's performance. Nevertheless, as the weight of $\sigma$ increases, the performance of backbones generally declines, resulting in the model adhering too strictly to the "established guidelines". Consequently, it impacts the efficacy of learning from the hard negative samples. We further use the density plots to visualize the prediction on the STS-Benchmark development set in Figure 7. These models are trained on the synthesized data without filtering. We can observe that in Figure 7 (a), the distribution of prediction results for labels $\geq 4$ is significantly shifted to the left. Compared with the results in Figure 7 (b), this issue is effectively alleviated, demonstrating the effectiveness of the Gaussian-decayed function in reducing the influence of false negative samples. To further verify the applicability of the Gaussian-decayed function, we applied it to SynCSE and verified the performance in Appendix E.

## 5 CONCLUSION

In this paper, we propose a pipeline-based data augmentation method using LLM to enhance data diversity in sentence representation learning. By leveraging knowledge of entities and quantities, our approach improves the model's ability to capture fine-grained semantic distinctions. The Gaussian-decayed function in our GCSE model further reduces noise in the generated data. Extensive experiments on STS and reranking tasks show that our method achieves state-of-the-art results with fewer synthesized samples and a more lightweight LLM, demonstrating its effectiveness and efficiency.

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

## APPENDIX

## A    DATA SYNTHESIS PROMPTS

In this section, we provide the specifics of our prompts for knowledge extraction and integration, and data synthesis. The particular prompts are presented in Table 6.

## B    VISUALIZATION OF SYNTHETIC SAMPLE DISTRIBUTION

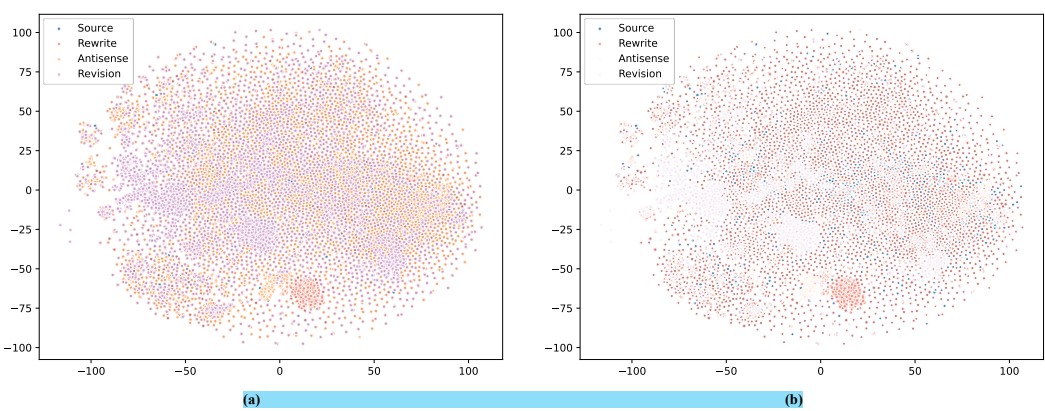

Figure 8: t-SNE visualization of the synthetic sample generated by ChatGLM3-6B, where the transparency of "Antisense" and "Revision" samples in subgraph (b) is reduced to 10% for better observation.

In this section, we use the supervised SimCSE model to generate sentence embeddings for the synthesized samples and utilize t-SNE to project the vectors into two-dimensional space for a visual analysis of the diversity. To facilitate observation, we group the synthesized samples into three categories: "Rewrite" refers to positive samples synthesized using "Rewriting Prompt 1" and "Rewriting Prompt 2" from Table 6, while "Antisense" denotes the negative samples generated using "Syntactic Antisense Prompt". "Revision" denotes the negative samples generated using "Entity Revision Prompt", "Quantity Revision Prompt" and "Rewriting Prompt 3", which are related to knowledge modification. And "Source" indicates the original samples from the dataset. We randomly selected 5k "Source" samples and corresponding synthetic samples from our dataset for visualization, and the results are illustrated in Figure 8. We observe that "Rewrite" samples basically cover the spatial distribution of "Source" samples while expanding into the neighborhood space to some extent. "Antisense" and "Revision" samples further enhance the information density within the target semantic space. Comparing Figure 8 (a) and (b), it can be observed that the "Revision" samples cover areas with sparse information, while their overall spatial distribution remains consistent with the semantic distribution of 'Source" samples. This indicates that the sample synthesis with knowledge effectively increases sample diversity within the semantic space.

## C    PERFORMANCE ON TRANSFER TASKS

We also evaluate our GCSE following the same settings as SimCSE on seven transfer tasks: MR (Pang & Lee, 2005), CR (Hu & Liu, 2004), SUBJ (Pang & Lee, 2004), MPQA (Wiebe et al., 2005),

| | |
|---|---|
| **Knowledge Extraction Prompt** | **Instruction:** Predicts the subject categories, contained entities, and quantified information of the following text 
 **Rules:** The category is an item in [{$categories\_name$}, ...], quantified information refers to information contained in the text with numerical values or units, such as '2GB', 'three cups', 'two dogs', etc 
 Output format: json format data, the data format is: { 
 cls: [], // category 
 entities: [{text: "", type: ""}], // entities, 'text' must be subsequences in the Input text 
 quantities: [{text: "", type: "", quantity: 0}] // To quantify the information, 'text' must be a subsequence in the Input text 
 } 
 **Input:** {$x$} |
| **Rewriting Prompt 1** | **Instruction:** You are an excellent storyteller; rewrite the input sentence in a different way. Please try to recreate the sentence using different expressions, including varied tones, synonyms, and sentence patterns, while ensuring that the new sentence has the same meaning as the original sentence. 
 **Input:** {$x$} |
| **Rewriting Prompt 2** | **Instruction:** You are a great storyteller; I would be grateful if you could employ your creativity to devise an illustration of the preceding segment of the sentence. The preceding statement must not exceed {$number$} words, and it follows the original text. 
 **Input:** {$x$} |
| **Rewriting Prompt 3** | **Instruction:** You are a great rewriter, and I want you to generate new sentence according to the classification, entities and quantities info provided by the json. 
 **Rules:** You should aware that the new text in "quantities" should be rewrite follows the "quantity" value. e.g. "text": "A man", "quantity": 5 should rewrite as "five men". 
 **Metadata:** { 
 "cls": "{$categories\_name$}", 
 "entities": [{ 

 "text": "{$entity\_text$}", 
 "type": "{$entity\_type$}" 
 }, ...], 
 "quantities": [{ 
 "text": "{$entity\_text$}", 
 "quantity": {$entity\_quantity$} 
 }, ...] } 
 **Input:** {$x$} |
| **Syntactic Antisense Prompt** | **Instruction:** You are dishonest; you ought to reformulate the input sentence so that the NLI model perceives it as an opposing sample. 
 **Rules:** 1. If the statement asserts negation, you should affirm; conversely, if the statement asserts affirmation, you should negate. 2. If an individual loves something, one should assert that it does not reciprocate that affection. 3. If an individual is engaged in one activity, state that they are performing a different activity. 4. If the statement is affirmative/negative, express it as negative/affirmative. 
 **Input:** {$x$} |
| **Entity Revision Prompt** | **Instruction:** You are a great story teller, rewrites the input sentence, and change the entity '{$original\_entity\_text$}' to another {$entity\_type$} '{$new\_entity\_text$}'. 
 **Input:** {$x$} |
| **Quantity Revision Prompt** | **Instruction:** You are a great story teller, rewrites the input sentence, and change the quantity {$original\_quantity\_value$} of '{$original\_quantity\_text$}' to {$random\_quantity\_value$}. 
 **Input:** {$x$} |

Table 6: Examples of data synthesis prompts, where {$variable\ name$} refers to a variable.

SST2 (Socher et al., 2013), TREC (Voorhees & Tice, 2000), and MRPC (Voorhees & Tice, 2000). The results are shown in Table 7, it can be observed that our GCSE (ChatGPT) achieves the best performance on all backbone models, outperforming second-best methods in average scores of 0.89% with BERT-base, 0.79% with BERT-large, 0.44% with RoBERTa-base, and 0.40% with RoBERTa-large, demonstrating the potential capability in downstream tasks.

| Model | Method | MR | CR | SUBJ | MPQA | SST2 | TREC | MRPC | Avg. |
|---|---|---|---|---|---|---|---|---|---|
| BERT-base | SimCSE♠ | 68.40 | 82.41 | 74.38 | 80.91 | 78.56 | 76.85 | 72.23 | 76.25 |
| | DiffCSE♠ | 72.28 | 84.43 | 76.47 | 83.90 | 80.54 | 80.59 | 71.23 | 78.49 |
| | PCL♠ | 72.84 | 83.81 | 76.52 | 83.06 | 79.32 | 80.01 | 73.38 | 78.42 |
| | RankCSE♠ | 75.66 | 86.27 | 77.81 | 84.74 | 81.10 | 81.80 | 75.13 | 80.36 |
| | MultiCSR (ChatGPT)♣ | 82.70 | 88.15 | 94.97 | 90.08 | 86.87 | 87.70 | 75.46 | 86.56 |
| | SynCSE (ChatGPT)* | 83.34 | 88.80 | 93.88 | 90.39 | 88.96 | 83.60 | 75.94 | 86.42 |
| | **GCSE (ChatGLM3-6B)** | 82.22 | 88.43 | 94.59 | 90.09 | 86.88 | **89.40** | 76.06 | 86.81 |
| | **GCSE (GLM4-9B)** | **84.63** | 89.78 | **95.01** | 90.54 | 88.96 | 86.00 | 76.12 | 87.29 |
| | **GCSE (ChatGPT)** | 84.59 | **90.15** | 94.97 | 90.39 | **89.68** | 86.00 | **76.35** | **87.45** |
| BERT-large | SimCSE♠ | 70.88 | 84.16 | 76.43 | 84.50 | 79.76 | 79.26 | 73.88 | 78.41 |
| | PCL♠ | 74.87 | 86.11 | 78.29 | 85.65 | 80.52 | 81.62 | 73.94 | 80.14 |
| | RankCSE♠ | 75.48 | 86.50 | 78.60 | 85.45 | 81.09 | 81.58 | 75.53 | 80.60 |
| | SynCSE (ChatGPT)* | 85.78 | 90.47 | 94.77 | 90.41 | 90.50 | 89.00 | **75.77** | 88.10 |
| | **GCSE (ChatGLM3-6B)** | 83.97 | 89.38 | 95.13 | 90.22 | 89.57 | 90.60 | 75.71 | 87.80 |
| | **GCSE (GLM4-9B)** | **86.01** | 90.94 | **95.40** | 90.24 | **92.15** | 92.00 | 75.48 | **88.89** |
| | **GCSE (ChatGPT)** | 85.93 | 90.44 | 94.94 | **90.52** | 92.04 | 88.80 | 75.25 | 88.27 |
| RoBERTa-base | SimCSE♠ | 70.16 | 81.77 | 73.24 | 81.36 | 80.65 | 80.22 | 68.56 | 76.57 |
| | DiffCSE♠ | 70.05 | 83.43 | 75.49 | 82.81 | 82.12 | 82.38 | 71.19 | 78.21 |
| | PCL♠ | 71.13 | 82.38 | 75.40 | 83.07 | 81.98 | 81.63 | 69.72 | 77.90 |
| | RankCSE♠ | 73.20 | 85.95 | 77.17 | 84.82 | 82.58 | 83.08 | 71.88 | 79.81 |
| | MultiCSR (ChatGPT)♣ | 84.70 | 90.69 | 94.40 | 89.38 | 89.42 | **89.62** | **77.01** | 87.89 |
| | SynCSE (ChatGPT)†† | 85.47 | 91.44 | 92.53 | 89.67 | 90.94 | 81.60 | 76.06 | 86.82 |
| | **GCSE (ChatGLM3-6B)** | 84.39 | 90.81 | 94.02 | 88.90 | 91.05 | 89.40 | 76.12 | 87.81 |
| | **GCSE (GLM4-9B)** | **86.49** | **92.24** | **94.70** | 89.63 | 92.37 | 86.60 | 76.29 | **88.33** |
| | **GCSE (ChatGPT)** | 86.32 | 91.58 | 94.37 | **90.04** | **92.42** | 84.00 | 76.12 | 87.84 |
| RoBERTa-large | SimCSE♠ | 72.86 | 83.99 | 75.62 | 84.77 | 81.80 | 81.98 | 71.26 | 78.90 |
| | PCL♠ | 74.08 | 84.36 | 76.42 | 85.49 | 81.76 | 82.79 | 71.51 | 79.49 |
| | RankCSE♠ | 73.20 | 85.83 | 78.00 | 85.63 | 82.67 | 84.19 | 73.64 | 80.45 |
| | SynCSE (ChatGPT)†† | 87.24 | **92.16** | 93.75 | **90.81** | 91.87 | 84.00 | **76.29** | 88.02 |
| | **GCSE (ChatGLM3-6B)** | 85.65 | 90.78 | 94.16 | 90.08 | 90.44 | **92.80** | 73.74 | 88.24 |
| | **GCSE (GLM4-9B)** | 87.45 | 91.60 | **94.62** | 90.30 | 92.42 | 88.40 | 71.77 | 88.08 |
| | **GCSE (ChatGPT)** | **87.56** | 91.76 | 94.56 | 90.69 | 92.26 | 88.80 | 74.84 | **88.64** |

Table 7: Comparison of different sentence embedding models accuracy on transfer tasks. "♠": results from Liu et al. (2023), "♣": results from Wang et al. (2024a), "††": results from Zhang et al. (2023). "*": we reproduce the results with the officially released corpus from Zhang et al. (2023).

| Premise | Hypothesis | Gold | SimCSE | RankCSE | SynCSE | GCSE |
|---|---|---|---|---|---|---|
| A woman is cooking eggs . | A woman is cooking something . | 3.00 | 4.37 (1.372) | 4.23 (1.320) | 3.66 (0.662) | **3.24 (0.236)** |
| Two little girls are talking on the phone. | A little girl is walking down the street. | 0.50 | 3.38 (2.881) | 3.64 (3.139) | 1.97 (1.468) | **1.85 (1.351)** |
| A chef is preparing some food . | A chef prepared a meal . | 4.00 | **4.27 (0.270)** | 4.59 (0.588) | 4.56 (0.561) | 4.41 (0.408) |
| Five kittens are eating out of five dishes . | Kittens are eating food on trays. | 2.75 | 3.81 (1.056) | 3.71 (0.957) | 3.28 (0.535) | **3.12 (0.373)** |
| A woman is cutting some herbs . | A woman is chopping cilantro . | 2.80 | 3.58 (0.777) | 3.58 (0.967) | 3.11 (0.313) | **2.61 (0.185)** |

Table 8: Case studies on model prediction similarity with gold labels in the STS-Benchmark development set, where Gold represents the label score of the sentence pair (ranging from zero to five). The similarity scores of all models are multiplied by a coefficient of five for better comparison, and the value in parentheses denotes the RMS error between the predicted score and the label. Words highlighted in blue denote the entity alteration in the sentence-pair, whereas words in yellow indicate the quantities that change inside the sentence-pair.

# D    CASE STUDIES

To further verify the improvement in our method's awareness of entity and quantity, we selected five sample sets from the STS-Benchmark development set that explicitly contained alterations in entity or quantity within the sentence-pair, and presented the prediction cosine-similarity scores of GCSE and related methodologies with the backbone of BERT-base in Table 8. We can observe from the results that the prediction score of our model achieves the minimum root-mean-square error compared to the label in most cases, which indicates that our model has a stronger capacity to distinguish information.

# E    ABLATION STUDIES OF GAUSSIAN-DECAYED AND FEW-SHOT SAMPLES

| Method | STS-12 | STS-13 | STS-14 | STS-15 | STS-16 | STS-B | SICK-R | Avg. |
|---|---|---|---|---|---|---|---|---|
| SynCSE (ChatGPT)* | 75.86 | 82.19 | 78.71 | 85.63 | **81.11** | 82.35 | **78.79** | 80.66 |
|   w sampled | 75.48 | 85.60 | 78.76 | 84.78 | 80.38 | 82.12 | 76.46 | 80.51 |
|   w sampled & G.D. | 75.71 | 85.24 | 79.09 | 85.15 | 80.82 | 82.68 | 77.54 | 80.89 |
|   w G.D. | **75.89** | 85.26 | 79.24 | 85.67 | 80.79 | 82.63 | 78.19 | **81.10** |
|   w sampled & domain & G.D. | 75.88 | **86.02** | **79.46** | **86.10** | 80.27 | **82.87** | 76.91 | 81.07 |

Table 9: Ablation studies of sample size and the Gaussian-decayed function by utilizing SynCSE. "*": we reproduce the results with the officially released corpus from Zhang et al. (2023).

We employ the Gaussian-decayed function on SynCSE and sample SynCSE training data with a sample size the same as our synthetic data to evaluate the efficacy of the proposed Gaussian-decayed function and our domain-oriented selection strategy in the ablation experiment. The data sample size is 64k, and the weight of $\sigma$ in $G(\cdot)$ is assigned the same value as specified in Section 4.1. The results of various policies implemented in SynCSE are presented in Table 9. "w sampled" denotes the utilization of purely the sampled data in SynCSE, and a performance decrease can be observed when training on a reduced number of samples without extra configurations. "w sampled & G.D." denotes the additional incorporation of $G(\cdot)$ based on "w sampled". "w G.D." indicates the results by training on the full dataset utilizing $G(\cdot)$. In both configurations, the average performance outperforms the vanilla model, illustrating the module's efficacy. "w sampled & domain & G.D." denotes the concurrent utilization of sample data, domain data, and $G(\cdot)$, with a sample size of 48k for the SynCSE dataset and 16k for the synthesized domain dataset. The results reveal that "w sampled & domain & G.D." attains the second-best performance, suggesting that incorporating domain data can decrease the required training samples while enhancing model efficacy.

# F    UNSUPERVISED SENTENCE EMBEDDING ON LLM

| Model | Avg. | Model | Avg. |
|---|---|---|---|
| *Unsupervised* | | *Data Augmentation* | |
| Llama3.2-3B LoRA | 71.34 | Llama3.2-3B LoRA | 78.26 |
| Llama-3-8B LoRA | **72.73** | Llama-3-8B LoRA | 78.24 |
| ChatGLM3-6B LoRA | 69.38 | ChatGLM3-6B LoRA | 79.04 |
| GLM4-9B LoRA | 71.77 | GLM4-9B LoRA | **79.52** |
| Qwen2.5-14B LoRA | 68.49 | Qwen2.5-14B LoRA | 78.02 |

Table 10: Performance comparison of different LLMs on STS tasks, where results of "Unsupervised" refers to models trained on the same unsupervised settings as Gao et al. (2021), and "Data Augmentation" refers to models trained with the synthetic data generated by ChatGLM3-6B.

In this section, we utilize contrastive learning on multiple LLMs to evaluate the alignment of LLM-generated similarities with the gold labels and the effectiveness of our data augmentation strategy. We use Llama3.2-3B (Dubey et al., 2024), Llama3-8B (Dubey et al., 2024), ChatGLM3-6B (GLM et al., 2024), GLM4-9B (GLM et al., 2024) and Qwen2.5-14B (Team, 2024; Yang et al., 2024) with a low-rank adapter (LoRA) layer for training. The sentence embedding vectors are obtained from the output hidden states of the last position, which is followed by the method of pretended chain of thought (Pretended CoT) (Zhang et al., 2024). We may derive two major conclusion from the results in Table 10: (1) In conventional unsupervised settings, decoder-based LLMs have no significant performance advantage over encoder-based PLMs for sentence representation learning tasks. The model performance does not increase significantly with the increase of the number of model parameters. To reduce expenses, we assert that fully leveraging the capabilities of LLMs for distilling smaller models is the better option. (2) The application of our data augmentation technique to sentence representation learning tasks in LLMs significantly enhances performance relative to the "Unsupervised" settings, which further proves the applicability and efficacy of our strategy.

# G    Visualization of Prediction Scores and Gradient Comparisons

Figure 9: Heatmap visualization of the prediction scores and gradients.

To further analyze the effectiveness of the Gaussian-decayed function in mitigating the impact of false negative noise, we visualized the changes in predicted scores and gradients during the training process using heatmaps. In the training procedure of GCSE, each input consists of a source sample, its corresponding positive sample, and a hard negative sample. We visualize the cosine similarity scores and gradient heatmaps for negative samples within a batch in Figure 9. Each cell of a heatmap represents the relationship between the source sample and the negative sample, and the diagonal cells highlight the relationships between source samples and their hard negatives. Since synthetic samples lack manual annotations, we use supervised SimCSE models (Gao et al., 2021) based on different backbones to compute their similarity scores as the ground truth. We normalized the output scores of each model with min-max scaling and averaged them as the final scores to address distributional differences across models, and the results are shown in Figure 9 (a-1). It can be observed that several hard negatives on the diagonal display scores biased towards positive similarity, indicating the presence of false negative noise. In the framework of contrastive learning, when optimized using standard contrastive loss, these hard negatives are positioned further from the source samples in the semantic space, negatively impacting the model's representational capacity. Figure 9 (a-2) displays the normalized cosine similarity scores of hard negatives in the initial step as calculated by the evaluation model in GCSE. The initial score distribution of hard negatives shows a strong correlation with the ground truth, suggesting that these scores could efficiently guide GCSE in gradient correction.

Figures 9 (b-1) and (b-2) present the backward gradient values of the model trained without and with the Gaussian-decayed function, respectively. For better visualization, all gradient values are amplified by $10^4$, and all similarities are amplified by 20 by the temperature. By comparing the gradients of hard negative samples in these two figures, it can be observed that the gradient values on false hard negatives are significantly smaller when the Gaussian-decayed function is applied. Additionally, Figures 9 (c-1) and (c-2) present a comparison of cosine similarity scores after 125 training steps with and without the Gaussian-decayed function. The scores for false hard negatives are significantly higher when the Gaussian-decayed function is employed, while the true hard negatives had lower scores. The overall score distribution aligns more accurately with the ground truth, and these results demonstrate that the Gaussian-decayed function effectively prevents false negatives from being pushed farther away from source samples in the semantic space, thereby validating its effectiveness in mitigating noise and improving model performance.

# H ABLATION ANALYSIS OF FILTERING THRESHOLDS

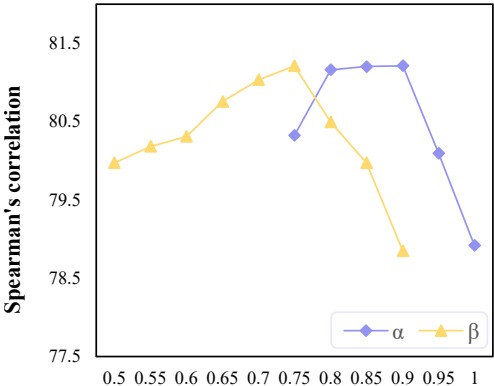

Figure 10: Spearman's correlation against the weight of $\alpha$ and $\beta$ on the STS tasks. When adjusting the weight of one parameter, the other parameter is fixed at its default value as specified in the experimental settings.

To study the impact of different filtering thresholds, we evaluate the performance on the backbone of the BERT-base, and the results are shown in Figure 10. When $\alpha > 0.9$, the model's performance declines significantly, primarily because the high threshold filters out too many samples, heavily reducing the number of positive samples. In the range $\alpha \in [0.8, 0.9]$, performance degradation is observed due to noise introduced by false positive samples. Similarly, when $\alpha < 0.8$, the model suffers from a performance drop caused by an excessive number of false positives being included in the training process. The threshold for $\beta$ demonstrates a noticeable impact on model performance when it deviates from 0.75. Specifically, when $\beta > 0.75$, the model's performance declines significantly due to the inclusion of excessive false negative noise, which severely affects the model performance. Conversely, when $\beta < 0.75$, the selected negative samples become easier for the model to distinguish, providing limited benefit for enhancing its representation learning capacity. The results highlight the influence of filtering thresholds on sample quality and distribution.

# I SCORE NORMALIZATION METHODOLOGY

In this work, the labels in datasets are normalized with standard min-max normalization. To address the discrepancy in score distributions among different models, we applied a variant min-max normalization method to align their predicted scores. For each label $l \in [0, \text{MAX}]$, we collect all predicted scores with $l = 0$ as list $C_0$, and all predicted scores with $l = \text{MAX}$ as list $C_1$. Specifically, we computed the median prediction scores for $C_0$ and $C_1$ as $min_p = \text{median}(C_0)$ and $max_p = \text{median}(C_1)$, respectively. The use of medians, rather than the minimum predicted score for $C_0$ or the maximum predicted score for $C_1$, avoids reliance on outlier values that may disproportionately skew the normalization, ensuring a more balanced score distribution. For a given score $s$, the normalized score $s'$ is calculated as:

$$s' = \text{clip}\left(\frac{s - min_p}{max_p - min_p}, 0, 1\right), \tag{14}$$

where the function $\text{clip}(x, 0, 1)$ ensures the normalized score is bounded within $[0, 1]$. This method adjusts the score range to maintain consistency across models while preserving relative score differences.

