# OpenReview forum: "Enhancing Unsupervised Sentence Embeddings via Knowledge-Driven Data Augmentation and Gaussian-Decayed Contrastive Learning"
_ICLR.cc/2025/Conference — Submitted to ICLR 2025_

### Official Review · Reviewer_Zenj · 2024-10-20

**Soundness:** 1
**Presentation:** 3
**Contribution:** 1
**Rating:** 5
**Confidence:** 4

**Summary:**

This paper proposes a pipeline-based data augmentation method via LLMs and introduces the Gaussian-decayed gradient-assisted Contrastive Sentence Embedding (GCSE) model to enhance unsupervised sentence embeddings. The data augmentation first constructs a knowledge graph (KG) with the extracted data, guides LLM to generate more diverse positive samples, and filter noisy data. Experimental results show its effectiveness.

**Strengths:**

1. The paper is well-written.
2. The data augmentation is interesting.

**Weaknesses:**

1. The main results are based on BERT and Roberta. However, LLMs are effective in sentence embedding [1,2]. Please use LLMs for evaluation. Using only ChatGLM-3 for evaluation is not convincing. Therefore, the experimental evaluation in this paper should be redone.

2. Analysis of \alpha and \beta.

3. Two missing references. [3] is about data noise and [4] is about data augmentation.

[1] BeLLM: Backward Dependency Enhanced Large Language Model for Sentence Embeddings, NAACL 2024

[2] AoE: Angle-optimized Embeddings for Semantic Textual Similarity, ACL 2024

[3] Debiased Contrastive Learning of Unsupervised Sentence Representations, ACL 2022

[4] Improving Text Embeddings with Large Language Models, ACL 2024

**Questions:**

Why the Gaussian-decayed function G() in the denominator? It seems that the Gaussian-decayed function has no connection with the denominator. Can we just add the Gaussian-decayed function to the contrastive learning loss?

---

> ### Author Response · Authors · 2024-11-20
>
> Thanks very much for taking your time to review the manuscript so seriously and responsibly. We sincerely apologize for our careless syntactic errors, and we will fix them in the revision. Please see the following for our response to your questions:
>
> **Q1: Why the Gaussian-decayed function G() in the denominator? It seems that the Gaussian-decayed function has no connection with the denominator. Can we just add the Gaussian-decayed function to the contrastive learning loss?**
>
> **A1:** Thank you for your question. In terms of the loss function, the Gaussian-decayed function G() could, in principle, be separated and directly incorporated into the contrastive learning loss. However, including it in the denominator simplifies the overall expression, preventing the formula from becoming unnecessarily complex.
>
> **Q2: The main results are based on BERT and Roberta. However, LLMs are effective in sentence embedding [1,2]. Please use LLMs for evaluation. Using only ChatGLM-3 for evaluation is not convincing. Therefore, the experimental evaluation in this paper should be redone.**
>
> **A2:** Thank you for your valuable feedback. We have expanded our experimental evaluation to include multiple LLMs for unsupervised sentence embedding tasks. Specifically, in **Appendix F**, we have added results for Llama3.2-3B, Llama3-8B, GLM4-9B, and Qwen2.5-14B, alongside ChatGLM3-6B, to assess the effectiveness of our data augmentation strategy across various models. The results are presented in **Table 10**.
>
> From these additional experiments, we can confirm that our data augmentation method is effective across different LLMs. However, it is important to note that while some LLM-based methods have achieved impressive results on supervised datasets, LLMs do not show a significant advantage over BERT-base in the unsupervised scenario.
>
> Moreover, our target scenario focuses on leveraging LLM knowledge to enhance the sentence representation learning capability of smaller encoder-based models. These models offer advantages in computational efficiency and resource requirements, enabling them to assist LLMs in tasks such as **RAG** (retrieval-augmented generation) for semantic querying. We hope these additional results and clarifications could address your concerns.
>
> **Q3: Analysis of \alpha and \beta.**
>
> **A3:** Thank you for your question. We have conducted an ablation analysis of the hyperparameters α and β in Equations 10 and 11. The impact of varying these hyperparameters on model performance is detailed in **Appendix H**, with the corresponding results presented in **Figure 10**.
>
> **Q4: Two missing references. [3] is about data noise and [4] is about data augmentation.** ** **
>
> **A4:** Thank you for pointing that out. We have added the missing references in the **Introduction** and **Related Work** sections to properly cite the relevant literature.

---

> > ### Comment · Reviewer_Zenj · 2024-11-20
> >
> > Thanks for your response. I raise my score to 5.
> >
> > The authors may consider the following questions to further enhance the quality.
> >
> > 1. The motivation of Gaussian-decayed function needs to be demonstrated more clearly.
> >
> > 2. It is not clear that the methods in this paper are superior to recent methods [1,2]. Furthermore, it is unclear whether the data augmentation method in this paper has an advantage compared to [3]. Conducting a thorough comparison and evaluation with recent related works would improve the soundness of this paper.
> >
> > [1] BeLLM: Backward Dependency Enhanced Large Language Model for Sentence Embeddings, NAACL 2024
> >
> > [2] AoE: Angle-optimized Embeddings for Semantic Textual Similarity, ACL 2024
> >
> > [3] Improving Text Embeddings with Large Language Models, ACL 2024

---

> > > ### Author Response · Authors · 2024-11-24
> > >
> > > **Q1: The motivation of Gaussian-decayed function needs to be demonstrated more clearly.**
> > >
> > > **A1:** Thanks for your comment, the evaluation model in our method is insufficient for effectively filtering false negatives with similar surface-level semantics in LLM-synthesized samples. To address this, we aim to balance sample diversity while mitigating the noise impact from false negatives. And that’s the motivation for proposing the Gaussian-decayed function. It plays a pivotal role in aligning hard negatives with the evaluation model's distribution during the initial training stage. This alignment ensures that false negatives are not erroneously pushed further apart in the semantic space.
> > >
> > > The Gaussian-decayed function serves two critical purposes in our GCSE model:
> > >
> > > - Gradient Regulation: Initially, it reduces the gradients of hard negatives, limiting the negative impact of noisy samples. As training progresses, the function allows gradient weights for hard negatives that diverge significantly from the evaluation model's distribution to recover, refining their placement in the semantic space.
> > >
> > >
> > > - Uniform Distribution: By carefully managing gradient magnitudes, it promotes a more uniform distribution of samples in the semantic space, preventing the amplification of noise-induced inconsistencies.
> > > To provide more concrete evidence, we have included a detailed analysis in **Appendix G**, which includes visualizations of the changes in the spatial distribution of negative samples. These results illustrate the effectiveness of the Gaussian-decayed function in mitigating noise and improving the quality of the learned semantic space.
> > >
> > > **Q2: It is not clear that the methods in this paper are superior to recent methods [1,2]. Furthermore, it is unclear whether the data augmentation method in this paper has an advantage compared to [3]. Conducting a thorough comparison and evaluation with recent related works would improve the soundness of this paper.**
> > >
> > > **A2:**
> > > Thank you for bringing up these recent works. While we acknowledge the significance of [1], [2], and [3], it is important to note that these methods are predominantly trained on **supervised NLI datasets**, whereas our approach relies solely on **unlabeled data**. Our method synthesizes on the unlabeled data via LLMs, and refined labels are further annotated using our evaluation model.
> > >
> > > To provide a fair comparison with [1] and [2], we conducted experiments using their publicly available code on our dataset synthesized with ChatGLM3. Following their default settings, we obtained the following average performance on the STS Benchmark:
> > >
> > > - AoE (BERT-based): 80.12
> > > - AoE (Llama2-7B): 18.14
> > > - BeLLM (Llama2-7B): 37.50
> > > - Native Llama2-7B: 78.58
> > >
> > > These results reveal that AoE (BERT-based) achieves good performance improvements compared to SimCSE, yet it does not consistently surpass other strong baselines (like RankCSE or SynCSE). Additionally, the performance of AoE and BeLLM with Llama2-7B appears unexpectedly low, contrasting with the stable performance of native Llama2-7B. We plan to investigate this further. Since there are no fundamental conflicts between these methods and ours, we may incorporate their results into future updates.
> > >
> > > Regarding [3], their data augmentation approach appears not to be open-sourced. We attempted to replicate it based on the prompts described in their paper, using ChatGLM3-6B for sample synthesis. This approach achieved an average Spearman score of 80.73, which is 0.48 points lower than our default method. However, due to the lack of access to their original implementation, our reproduction may not be entirely accurate, and we may not include these results in our manuscript.
> > >
> > > We appreciate your suggestion and will continue refining our comparisons to enhance the soundness of the paper.

---

### Official Review · Reviewer_wcpf · 2024-10-27

**Soundness:** 2
**Presentation:** 2
**Contribution:** 2
**Rating:** 3
**Confidence:** 5

**Summary:**

This paper proposes a data augmentation method related to the STS task, aiming to prompt LLMs for entity extraction, knowledge graph construction, and data generation. Additionally, it introduces a Gaussian-decayed function to align hard negatives between a training encoder and a frozen encoder.

**Strengths:**

The proposed pipeline, which leverages prompt engineering, entity extraction, and knowledge graph construction, introduces a novel paradigm for data augmentation in STS tasks. The paper is well written and easy to follow.

**Weaknesses:**

1、The motivation is not clearly clarified. why was a Gaussian-decayed function chosen?
2、The experimental improvement is marginal.

missing references;
[1]  Zhou K, Zhang B, Zhao W X, et al. Debiased contrastive learning of unsupervised sentence representations[J]. arXiv preprint arXiv:2205.00656, 2022.
[2] Wu X, Gao C, Su Y, et al. Smoothed contrastive learning for unsupervised sentence embedding[J]. arXiv preprint arXiv:2109.04321, 2021.

**Questions:**

1、Why was a Gaussian-decayed function chosen?
2、Why train a frozen encoder first, followed by training a new encoder with LLM-generated data? What's the benefit?

---

> ### Author Response · Authors · 2024-11-20
>
> Thanks very much for taking your time to review the manuscript so seriously and responsibly. We have added the missing references, and please see the following for our response to your questions:
>
> **Q1: 1、Why was a Gaussian-decayed function chosen?**
>
> **A1:** Thank you for your question. We chose the Gaussian-decayed function to balance sample diversity while minimizing the impact of noise from false negatives during training. Specifically, in the initial training steps, the Gaussian-decayed function helps align all hard negatives with the distribution of the evaluation model. As training progresses, it allows gradient weights for hard negatives that deviate more significantly from the evaluation model's distribution to recover gradually.
>
> **Q2: 2、Why train a frozen encoder first, followed by training a new encoder with LLM-generated data? What's the benefit?**
>
> **A2:** Thank you for your question. The reason we first train a frozen encoder and then train a new encoder with LLM-generated data is to address the noise present in the synthetic data, particularly false positive and false negative samples. Due to the generative ability and potential biases of LLMs, it is difficult to eliminate this noise during the data synthesis process.
>
> To mitigate this issue, we utilize an unsupervised evaluation model trained on data from the same distribution, which helps the GCSE filter out some of the noise present in the generated samples. Furthermore, for hard negative samples that are more challenging to classify, we use the Gaussian-decayed function to align with the evaluation model’s probability distribution during the initial training phase. It reduces the loss for all hard negative samples and ensures that the model is not influenced by noisy hard negatives in the early stages of training. This two-step process helps the model achieve better performance by first creating a robust evaluation model that filters out noise, and then training the new encoder with more reliable guidance.

---

> > ### Author Response · Authors · 2024-11-24
> >
> > Thanks for your review. We would like to supplement two statements of our revision:
> >
> > 1. **The motivation of the Gaussian-decayed function statement:** the evaluation model in our method is insufficient for effectively filtering false negatives with similar surface-level semantics in LLM-synthesized samples. To address this, we aim to balance sample diversity while mitigating the noise impact from false negatives. And that’s the motivation for proposing the Gaussian-decayed function. It plays a pivotal role in aligning hard negatives with the evaluation model's distribution during the initial training stage. This alignment ensures that false negatives are not erroneously pushed further apart in the semantic space.
> >
> > The Gaussian-decayed function serves two critical purposes in our GCSE model:
> >
> > - Gradient Regulation: Initially, it reduces the gradients of hard negatives, limiting the negative impact of noisy samples. As training progresses, the function allows gradient weights for hard negatives that diverge significantly from the evaluation model's distribution to recover, refining their placement in the semantic space.
> >
> >
> > - Uniform Distribution: By carefully managing gradient magnitudes, it promotes a more uniform distribution of samples in the semantic space, preventing the amplification of noise-induced inconsistencies.
> > To provide more concrete evidence, we have included a detailed analysis in **Appendix G**, which includes visualizations of the changes in the spatial distribution of negative samples. These results illustrate the effectiveness of the Gaussian-decayed function in mitigating noise and improving the quality of the learned semantic space.
> >
> > 2. **The concern about the marginal model performance:** We have conducted additional experiments using the same experimental settings but synthesized samples with GLM4-9B and ChatGPT. The results of these experiments have been added to Tables 2 and 7.
> > Notably, our model demonstrates substantial improvements over prior best methods. For semantic textual similarity (STS) tasks, our approach outperforms the previous state-of-the-art by:
> >
> > - 1.05% with BERT-base,
> > - 1.62% with BERT-large,
> > - 0.49% with RoBERTa-base,
> > - 1.50% with RoBERTa-large.

---

> ### Author Response · Authors · 2024-11-27
>
> Dear Reviewer,
>
> We thank again for your contributions to the reviewing process.
> The responses to your concerns and the corresponding paper revision have been posted. Please let us know whether we have properly addressed your concerns. We look forward to your reply and welcome any further questions.
>
> Best regards,
>
> Authors of Paper: "Enhancing Unsupervised Sentence Embeddings via Knowledge-Driven Data Augmentation and Gaussian-Decayed Contrastive Learning"

---

> > ### Comment · Reviewer_wcpf · 2024-12-03
> >
> > Thanks for your reply. I will keep my score unchanged.

---

### Official Review · Reviewer_PZ9X · 2024-10-28

**Soundness:** 2
**Presentation:** 2
**Contribution:** 2
**Rating:** 3
**Confidence:** 4

**Summary:**

This paper proposes a STS task-related data augmentation method, prompting LLM for entity extraction, KG construction and data generation. It also introduces a Gaussian-decayed function to align hard negatives between a training encoder and a frozen encoder.

**Strengths:**

1. The paper is well-written. The figures are neat and clear to read.
2. The pipeline of utilizing prompt engineering, entity extraction and KG construction serves as a new paradigm of data augmentation for STS tasks.

**Weaknesses:**

1. Analysis of false positive (FP) and negative (FN) in Figure 1 is unconvincing. See Question 1 for details.
2. The improvements (0.28 with BERT-base, 0.23 with BERT-large, and 0.46 with RoBERTa-large) seem very marginal.
3. The motivation of choosing Gaussian-decayed function is unclear. No theoretical proves on the modification with Gaussian-decayed function.

**Questions:**

1. In figure 1, why choosing 0.5 as a hard limit to distinguish Positives & Negatives? Can we choose other limits, like >0.8 is Positives, < 0 is Negatives, and 0~0.8 is ambiguous? Theoretically,  most embedding models like SimCSE are trained with a list-wise temperature-scaled contrastive loss, which is basically optimized via Cross-Entropy. The distribution of query-positive cosine similarity scores will be pushed greatly towards 1.0, and query-negative scores towards -1.0. If SimCSE is trained with some optimization object like a point-wise MSE loss, choosing 0.5 will be more rational. Thus, I suggest you to set a hard limit Top-k window for the prediction of Positives & Negatives. For example, the prediction with largest similarity is positive, and lowest is negative.
2. Can you do statistic significant tests on your results?
3. Where is the evaluation model E′ in Figure 4?
4. Why choosing Gaussian-decayed function? Does it have some special mathematical traits? Can we just choose a MSE loss / Cross-Entropy loss / KL loss to align hard negatives between the training encoder and frozen encoder?
5. Why do you want to align the hard negatives between the training encoder and frozen encoder?
6. Why do you train a frozen encoder first, then train a new encoder with the LLM-generated data? Can we just merge these steps to one stage?
7. Line 311: How would the “spatial distribution of negatives” change? How will the gradients update?

---

> ### Author Response · Authors · 2024-11-20
>
> Thanks very much for taking your time to review the manuscript so carefully. Please see the following for our response to your questions:
>
> **Answer 1:** Thank you for your insightful suggestion! We acknowledge that the choice of the threshold 0.5 may not be universally optimal for distinguishing positive and negative samples. We have carefully considered your proposed approach of selecting positives and negatives based on the top-k similarity scores. While this method is effective for binary classification tasks, we believe it might not adequately capture the fine-grained distribution of sentence embedding similarity scores, which is critical for sentence representation learning tasks.
>
> To address this limitation, we introduced a new score normalization method to better align predicted scores with the target labels. Details of the normalization formula and its implementation have been provided in **Appendix I**. Following the normalization, we defined false negative samples as those with a root mean square error (RMSE) greater than 0.2 between predicted and normalized label scores. Positive samples were selected as those where the predicted score exceeds the corresponding label. We have updated **Figure 1** with the recalibrated predictions based on this improved approach.
>
> **Answer 2:** Thank you for your question. Could you please clarify which specific results you are referring to for statistical significance testing? In our main experiments, we followed prior work and used the SentEval for performance evaluation. We validated our approach on multiple benchmark datasets, and some experiments were conducted with k-fold cross-validation. This methodology is designed to mitigate the influence of randomness in the results.
>
> **Answer 3:** Thank you for pointing this out. We have updated **Figure 4** to include a clearer annotation within the Gaussian-Decay function module, explicitly indicating the role of E′.
>
> **Answer 4:** Thank you for your question. We chose the Gaussian-decayed function to balance sample diversity while minimizing the impact of noise from false negatives during training. Specifically, in the initial training steps, the Gaussian-decayed function helps align all hard negatives with the distribution of the evaluation model. As training progresses, it allows gradient weights for hard negatives that deviate more significantly from the evaluation model's distribution to recover gradually. This smooth adjustment is a key advantage over alternative loss functions like MSE or Cross-Entropy.
>
> We observed that using KL divergence loss exhibits instability in certain situations. This instability likely arises from the sparsity of the label distribution. For example, when probabilities for certain classes are close to zero or exactly zero, and the predicted probabilities approach one, the logarithmic term in KL divergence can become excessively large, leading to highly negative loss values. By contrast, the Gaussian-decayed function provides a more stable and effective way to achieve our training objectives.
>
> **Answer 5, 6:** Thank you for these questions. The alignment of hard negatives between the training encoder and the frozen evaluation encoder is motivated by the potential noise in LLM-generated data, which can include false positive and negative samples. Due to the generative ability and potential biases of LLMs, it is challenging to eliminate such noise during the data synthesis process.
>
> The evaluation encoder, trained on data from the same distribution, serves as a noise-filtering mechanism to initially reduce the impact of noisy samples in the generated data. Secondly, for the false negative noise in the hard negatives, which is more difficult to distinguish. We aim to align the training encoder with the evaluation encoder’s probability distribution during the initial training step.
>
> As for the two-stage training approach, it is necessary to first train the evaluation encoder in an unsupervised manner before using it to assist in training the new encoder with LLM-generated data. Combining these steps into a single stage would bypass the noise-filtering benefits provided by the evaluation encoder and could lead to suboptimal performance.
>
> **Answer 7:** Thank you for your question. Initially, the gradients for hard negatives are relatively small due to the Gaussian-decayed loss. However, as training progresses, the model leverages other in-batch negative pairs to refine the semantic space. This process gradually alters the predicted scores for hard negatives.
>
> For those hard negatives whose predicted scores increasingly diverge from the evaluation model's scores, the gradient decay applied to these samples progressively diminishes. We have further visualized and analyzed the changes in the spatial distribution of negatives in **Appendix G**, providing additional descriptions into this process.

---

> > ### Author Response · Authors · 2024-11-24
> > **A supplement to the Q2 response**
> >
> > We have conducted statistical significance tests on the main results in **Table 2**. We performed **t-tests** to evaluate the significance of the differences in average performance. The results indicate that **p < 0.5%**, demonstrating statistically significant differences between GCSE and the baselines.
> >
> > We have updated **Table 2** to include these significant test results.

---

> ### Author Response · Authors · 2024-11-27
>
> Dear Reviewer,
>
> We thank again for your contributions to the reviewing process.
> The responses to your concerns and the corresponding paper revision have been posted. Please let us know whether we have properly addressed your concerns. We look forward to your reply and welcome any further questions.
>
> Best regards,
>
> Authors of Paper: "Enhancing Unsupervised Sentence Embeddings via Knowledge-Driven Data Augmentation and Gaussian-Decayed Contrastive Learning"

---

### Official Review · Reviewer_Hrdn · 2024-10-31

**Soundness:** 3
**Presentation:** 3
**Contribution:** 3
**Rating:** 6
**Confidence:** 4

**Summary:**

To address the challenges of unsupervised sentence embedding models in terms of data diversity and noise, this paper first proposes a pipeline-based data augmentation method using large language models. This method utilizes knowledge graphs to extract entity and quantity information, generating more diverse samples. Next, the paper introduces a Gaussian Contrastive Sentence Embedding (GCSE) model assisted by a Gaussian decay function to reduce the impact of false negative samples and enhance the model's discriminative ability. Finally, the paper demonstrates that this approach achieves SOTA performance on semantic textual similarity  tasks, while exhibiting efficiency and robustness with fewer data samples and smaller LLMs.

**Strengths:**

1. By integrating knowledge graphs and LLMs, the paper proposes a novel data augmentation approach that effectively enhances sample diversity and embedding quality.
2. The introduction of the Gaussian decay function significantly mitigates the impact of false negative samples, bolstering the model's robustness.
3. While reducing data requirements and model size, the method presented in this paper achieves state-of-the-art performance across multiple benchmarks, demonstrating its efficiency and generalization capability.

**Weaknesses:**

1. The core motivation of this paper's approach is data diversity and data noise. Although higher benchmark performance has been achieved, there is a lack of significant quantitative analysis on how the proposed methods or modules enhance data diversity and reduce data noise. For example, could some quantitative metrics be used to intuitively demonstrate the improvements in data diversity and reductions in data noise brought about by the proposed method or individual modules?

**Questions:**

1. Is it possible to visually demonstrate the improvement in data diversity and reduction in data noise brought about by the proposed method through some quantitative metrics (same as Weakness) ?

2. Ablation analysis of the hyperparameters α and β in equations 10 and 11.

3. The impact of different LLMs on performance, and whether replacing with a more powerful LLM could raise the method's upper limit.

---

> ### Author Response · Authors · 2024-11-20
> **Author Response**
>
> Thanks very much for taking your time to review the manuscript so seriously and responsibly. Please see the following for our response to your questions:
>
>
> **Q1: 1. Is it possible to visually demonstrate the improvement in data diversity and reduction in data noise brought about by the proposed method through some quantitative metrics (same as Weakness) ?**
>
>
> **A1:** Thank you for your suggestion. We have supplemented the analysis with additional visualizations to demonstrate the improvements in data diversity and the reduction of data noise brought about by our proposed method.
> In **Appendix B**, we provide t-SNE visualizations of the synthesized samples. The results indicate that incorporating entity knowledge significantly enhances the diversity of samples in the semantic space.
> To address potential noise in the synthesized samples and the role of the Gaussian-decayed function in mitigating its impact, we have included further analyses in **Appendix G**. This section features visualizations and examines the changes in the spatial distribution of negative samples, highlighting the effectiveness of our approach in handling noisy data.
> We hope these additions provide the quantitative and visual clarity you are seeking.
>
>
> **Q2: 2. Ablation analysis of the hyperparameters  α and β in equations 10 and 11.**
>
>
> **A2:** Thank you for your question. We have conducted an ablation analysis of the hyperparameters α and β in Equations 10 and 11. The impact of varying these hyperparameters on model performance is detailed in **Appendix H**, with the corresponding results presented in **Figure 10**.
>
> **Q3: 3. The impact of different LLMs on performance, and whether replacing with a more powerful LLM could raise the method's upper limit.**
>
> **A3:** Thank you for raising this important question. The quality of synthetic samples indeed has a significant impact on the final model performance, and the parameter size of the LLM can influence both the adherence to prompts and the quality of the generated samples. Theoretically, using a larger LLM should result in better performance due to higher-quality sample synthesis. To explore this, we conducted additional experiments using the same experimental settings but synthesized samples with GLM4-9B and ChatGPT. The results of these experiments have been added to **Tables 2** and **7**. Notably, our model demonstrates substantial improvements over prior best methods. For semantic textual similarity (STS) tasks, our approach outperforms the previous state-of-the-art by:
> + 1.05% with BERT-base,
> + 1.62% with BERT-large,
> + 0.49% with RoBERTa-base,
> + 1.50% with RoBERTa-large.

---

> > ### Comment · Reviewer_Hrdn · 2024-11-25
> > **Response to authors**
> >
> > I appreciate the author's substantial work during the rebuttal period. I am willing to increase my confidence score from 3 to 4.

---

> > > ### Author Response · Authors · 2024-11-27
> > >
> > > Thank you for your thoughtful feedback and for reconsidering your evaluation. We truly appreciate the time and effort you invested during the rebuttal period.

---

### Official Review · Reviewer_n2jX · 2024-11-08

**Soundness:** 3
**Presentation:** 3
**Contribution:** 2
**Rating:** 5
**Confidence:** 4

**Summary:**

This paper investigates the use of large language models for data augmentation to generate unsupervised sentence embeddings. To overcome two limitations of existing methods: limited data diversity and high data noise, the authors propose using knowledge graphs to extract entities and quantities to address low data diversity. Additionally, they suggest using a Gaussian-decayed function to limit the impact of false hard negative samples in the Contrastive Sentence Embedding model to tackle the issue of high data noise. Experimental results on semantic textual similarity (STS) and reranking tasks indicate that the proposed method offers certain improvements over existing large language model-based approaches.

**Strengths:**

The authors have provided a thorough analysis and summary of the shortcomings of existing methods.

This paper is easy to follow, and the technical approach is sound.

**Weaknesses:**

Although introducing the Gaussian-decayed function into SimCSE is straightforward, its contribution does not seem very significant.

From the results in Tables 2 and 3, the improvement of GCSE over SynCSE appears to be relatively minor in many cases. I suggest conducting multiple experiments and reporting the results of statistical significance tests.

**Questions:**

If SynCSE and GCSE use the same LLM, would the improvement be greater? Could you provide these results?

---

> ### Author Response · Authors · 2024-11-20
>
> Thanks very much for taking your time to review this manuscript! Please see our response to your comments below:
>
>
> **Q1: If SynCSE and GCSE use the same LLM, would the improvement be greater? Could you provide these results?**
>
>
> **A1:** Thank you for raising this important question. The quality of synthetic samples indeed has a significant impact on the final model performance, and the parameter size of the LLM can influence both the adherence to prompts and the quality of the generated samples. Theoretically, using a larger LLM should result in better performance due to higher-quality sample synthesis.
> To explore this, we conducted additional experiments using the same experimental settings but synthesized samples with GLM4-9B and ChatGPT. The results of these experiments have been added to **Tables 2** and **7**.
>
> Notably, our model demonstrates substantial improvements over prior best methods. For semantic textual similarity (STS) tasks, our approach outperforms the previous state-of-the-art by:
> + 1.05% with BERT-base,
> + 1.62% with BERT-large,
> + 0.49% with RoBERTa-base,
> + 1.50% with RoBERTa-large.
>
> These results underscore the robustness of our method across different LLMs and model architectures.

---

> ### Author Response · Authors · 2024-11-27
>
> Dear Reviewer,
>
> We thank again for your contributions to the reviewing process.
> The responses to your concerns and the corresponding paper revision have been posted. Please let us know whether we have properly addressed your concerns. We look forward to your reply and welcome any further questions.
>
> Best regards,
>
> Authors of Paper: "Enhancing Unsupervised Sentence Embeddings via Knowledge-Driven Data Augmentation and Gaussian-Decayed Contrastive Learning"

---

### Author Response · Authors · 2024-11-20
**General Response to Reviewers**

Thanks very much for taking the time to review our manuscript, and we really appreciate all the comments and suggestions.
In response to the two major concerns regarding the motivation behind the Gaussian-decayed function and whether the model performance can achieve the upper limit with more powerful LLMs:
(1) We have expanded on the motivation of the Gaussian-decayed function in the Introduction to provide further clarification. Additionally, we have included a detailed analysis in Appendix G, where we visualize and demonstrate how the Gaussian-decayed function helps in mitigating the noise from negative samples.
(2) Regarding model performance, we extended our experiments using the same setup and evaluated the performance of our model with samples synthesized by GLM4-9B and ChatGPT. The results of these experiments have been added to Tables 2 and 7. Our model outperforms previous state-of-the-art methods in average scores for semantic textual similarity (STS) tasks, achieving improvements of:
	1.05% with BERT-base
	1.62% with BERT-large
	0.49% with RoBERTa-base
	1.50% with RoBERTa-large

---

### Meta-Review · Area_Chair_Gw8p · 2024-12-20

**Metareview:**

This paper introduces a pipeline-based data augmentation method utilizing LLMs combined with a novel Gaussian-decayed gradient-assisted Contrastive Sentence Embedding (model to enhance unsupervised sentence embeddings. The proposed approach integrates knowledge graphs for data diversity and employs a Gaussian decay function to mitigate data noise, resulting in good performance on semantic textual similarity tasks.

However, as pointed out by the reviewers, there are several concerns, including the marginal improvements over existing methods, the unclear theoretical justification for the Gaussian-decayed function, and the lack of significant quantitative analysis demonstrating how the proposed methods enhance data diversity or reduce noise. Even though the authors tried to address these issues during the rebuttal, the reviewers were not fully satisfied with the answers, particularly regarding the practical significance and theoretical underpinnings of the improvements reported.

**Additional Comments On Reviewer Discussion:**

Nil.

---

### Decision · Program_Chairs · 2025-01-22

Reject